# A genome-wide analysis of YY1 and TFAP2 competition on overlapping motifs reveals their roles in HPV-induced carcinogenesis

Yi Liu[1,2☉], Shuang Ding[1☉], Haibin Liu [1,2*]

**1** Center for Emerging Infectious Diseases, Wuhan Institute of Virology, Center for Biosafety Mega-Science, Chinese Academy of Sciences, Wuhan, China, **2** Hubei Jiangxia Laboratory, Wuhan, China

☉ These authors are contributed equally to this work.
\* hbliu@wh.iov.cn

## Abstract

The long non-coding RNA lnc-FANCI-2 acts as a host defense RNA and is highly expressed in HPV-positive cervical lesions. Its activation relies on the binding of the transcription factor YY1 to two conserved motifs in its promoter. We used DNA oligo pull-down combined with mass spectrometry to identify proteins binding to the lnc-FANCI-2 promoter, discovering new TFAP2 family members that compete with YY1 for binding at overlapping sites. In primary epithelial cells, TFAP2 binding led to lnc-FANCI-2 silencing. However, in HPV-positive cancer cells, increased YY1 levels displaced TFAP2, alleviating repression. Genome-wide predictions using the JASPAR database identified thousands of YY1 and TFAP2 competition binding sites (CBSs), many overlapping with CHIP-seq peaks for YY1, TFAP2A, and TFAP2C, predominantly in promoter regions. We validated competition at two CBSs in the promoter and found it likely regulates cancer-related genes PPP1R15B and LRRC37A. This suggests that YY1 and TFAP2 competition might influence a broader transcriptional regulation network in HPV-induced cancer. This study reveals a novel transcriptional antagonism mechanism affecting lnc-FANCI-2 and other cancer-related genes, highlighting YY1 and TFAP2 as potential therapeutic targets in HPV-driven carcinogenesis.

## Author summary

Cervical cancer, primarily caused by high-risk human papillomaviruses (HPV), remains an important global health threat. A key player in HPV-induced cancer is the long non-coding RNA lnc-FANCI-2, which is highly expressed in HPV-positive lesions and helps suppress tumor-promoting pathways. However, how its expression is regulated remains unclear.

**Data availability statement:** All data are in the manuscript and/or supporting information files.

**Funding:** This study was funded by National Key R&D Program of China (2023YFC2605103 to HL), National Natural Science Foundation of China (32470162 to HL), and Biosafety and Technology Major Program of Hubei Jiangxia Laboratory (No. JXBS016 to HL). The funders had no role in study design, data collection and analysis, decision to publish, or preparation of the manuscript.

**Competing interests:** The authors have declared that no competing interests exist.

In this study, we discovered that two transcription factors, YY1 and TFAP2, compete for binding at an overlapping site in the lnc-FANCI-2 promoter. In normal cells, TFAP2 silences lnc-FANCI-2, but in HPV-infected cells, elevated YY1 displaces TFAP2, activating lnc-FANCI-2. Using genome-wide analysis, we identified thousands of similar competition sites across the genome, many near cancer-related genes like PPP1R15B and LRRC37A, which are linked to poor survival in cervical cancer patients.

Our findings reveal a new transcriptional network between YY1 and TFAP2 that influences HPV-driven cancer. By understanding this competition, we may uncover new therapeutic strategies to disrupt harmful gene regulation in cervical cancer and other HPV-associated diseases.

## Introduction

Cervical cancer is the fourth most common cancer and leading cause of death among women globally, with approximately 604,000 new cases and 342,000 deaths annually [1,2]. Persistent infection with high-risk HPVs (hrHPVs), particularly HPV16 and HPV18, is a major driver of cervical carcinogenesis, contributing to the development of high-grade cervical intraepithelial lesions (CIN) and invasive cervical cancer (ICC) [3–5].

HPV E6 and E7 oncoproteins are persistently expressed from a single integration site within the host genome and are crucial for maintaining the transformed state of HPV-induced cancer cells [6]. Although E6 and E7 lack enzymatic activities, they interact with a number of host tumor suppressors, leading to abnormal transcriptional regulation [7–9]. E6 targets the tumor suppressor p53 to evade p53-mediated anti-tumor responses [10–13], whereas E7 promotes aberrant cell cycle progression by degrading pRb [14,15]. In addition, E6 and E7 interact with other transcription factors like c-myc, c-jun, IRF, p300, and HDAC [7–9], indicating their role in broader carcinogenic processes. Notably, HPV16 E7 has been shown to functionally interact with the transcriptional regulator Yin Yang 1 (YY1), modulating its activity at specific promoters [16]. However, E6 and E7 alone are insufficient to fully transform primary cells, suggesting that further genetic, epigenetic, or transcriptional changes are necessary for cervical cancer development [17]. Hence, understanding these additional transcriptional alterations —particularly those mediated through E7-YY1 interactions—is essential for comprehending HPV-induced carcinogenesis.

YY1 is a host transcription factor that can act as both activator and repressor, and its overexpression is linked to various cancers [18]. YY1 represses early HPV gene transcription through specific binding motifs in the long control region [19]. It is upregulated due to reduced miR-29a in HPV-positive cancer cells, leading to the activation of the cancer-associated long non-coding RNA lnc-FANCI-2 [16]. Two essential YY1 binding motifs upstream of lnc-FANCI-2's transcription start site (TSS) are crucial for its transcriptional activity [16] (Fig 1A). YY1

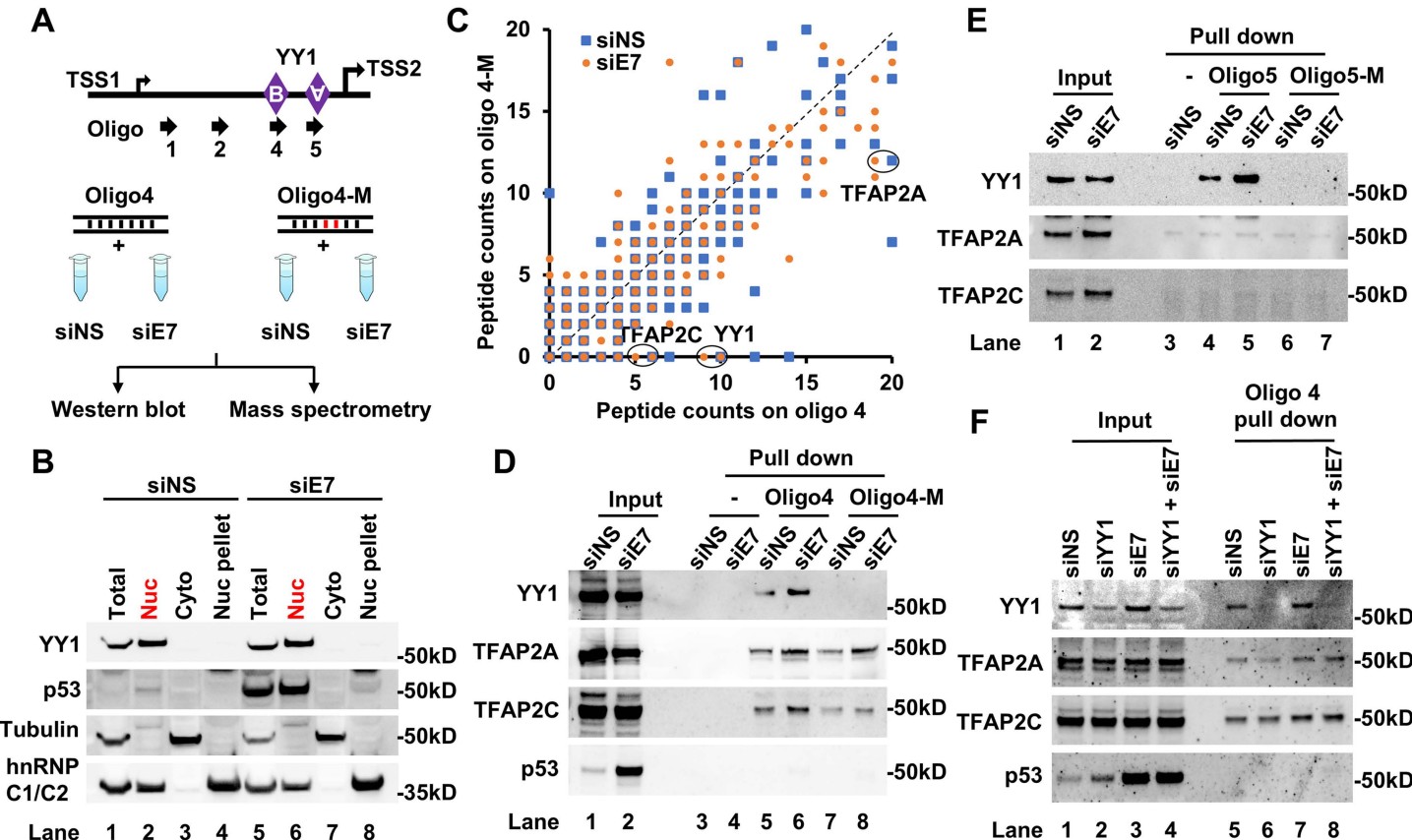

**Fig 1. TFAP2 preferentially binds to lnc-FANCI-2 promoter core region.** (A-C) Identification and validation of proteins binding to a YY1-binding motif B. The wild-type YY1 motif containing oligo 4 (oligo 4) and its mutant oligo 4 (oligo 4-M) were used for the pulldown assays with the nuclear extracts from CaSki cells with or without siRNA knockdown of HPV16 E7 expression (siE7, E7-specific siRNA; siNS, nonspecific siRNA control). (A) Workflow of DNA oligo pulldown and mass-spectrometry assays. (B) Knockdown (KD) of E7 in CaSki cells was performed as described previously [16]. The soluble nuclear extract (Nuc in red) was isolated by fractionation and used for DNA oligo pulldown assays in panels C-E. The KD efficiency of HPV16 E7 was indicated by stabilized and increased p53 protein level. Fractionation efficiency was evaluated by immunoblot of tubulin in the cytoplasm (Cyto) and hnRNP C1/C2 in the nucleus. (C) A scatter plot shows the peptide counts for proteins on oligo4 over that on oligo4-M which were identified by mass spectrometry. (D-F) TFAP2 binds to the YY1 motif B, but not the motif A. (D) Binding of TFAP2A and TFAP2C to the YY1 motif B. The oligo4 with YY1 motif B and oligo4-M with a mutant motif B were used for the pulldown and Western blot analyses using fractionated nuclear extracts isolated from the CaSki cells with indicated siRNA KD. (E) YY1 motif A in lnc-FANCI-2 promoter binds YY1, but not TFAP2A and TFAP2C. Pulldown assays of oligo5 containing YY1 motif A and its mutant oligo5-M were performed as described for oliog4 pulldown and Western blot analyses. (F) YY1 motif B binds to both TFAP2A and TFAP2C independent of YY1. The YY1-motif B containing Oligo 4 was used for the pulldown assays with soluble nuclear extracts fractionated from CaSki cells treated with non-specific siRNA (siNS), YY1 siRNA (siYY1) or E7 siRNA (siE7). KD efficiency was measured by stabilized p53 protein.

knockdown or motif deletion [20] causing a 70–80% reduction in lnc-FANCI-2 expression in cervical cancer cells. In addition, HPV16 E7 directly binds YY1 via its CR3 domain in a DNA-independent manner and enhances its trans-activator function on the lnc-FANCI-2 promoter, independently of YY1's acetylation status or interaction with p300. This E7-YY1 axis forms the foundational mechanism for the transcriptional upregulation of lnc-FANCI-2 upon HPV infection. This study aims to identify additional host proteins binding to the lnc-FANCI-2 promoter through proteomic analysis and proposes a model for how TFAP2 and YY1 regulate lnc-FANCI-2 and other cancer-related genes in both primary and cancer cells.

## Results

### TFAP2 proteins bound to lnc-FANCI-2 promoter core region independent of YY1 and HPV16 E7

Although two YY1-binding motifs, YY1-A and YY1-B (Fig 1A), in the lnc-FANCI-2 promoter region have very similar binding affinities to YY1, the YY1-B is more critical for the lnc-FANCI-2 promoter activity [17]. To comprehensively characterize how HPV16 E7 modulates the transcriptional complex at this locus, we performed DNA oligo pull-down followed by mass spectrometry using soluble nuclear extract (Nuc) from HPV16+ CaSki cells with or without E7 siRNA transfection (Fig 1A and 1B) [16]. This experimental design enabled the identification of baseline protein interactions at the lnc-FANCI-2 promoter core region and the distinction of E7-dependent binding partners via comparative analysis. Previous work demonstrated E7 siRNA also targeted E6 transcripts and resulted in the accumulation of nuclear p53 (Figs 1B and S1A) [21]. This p53 accumulation served as an indicator of effective E6/E7 knockdown. Using this approach, we identified 28 proteins specifically enriched in the oligo 4 pull-down compared to its mutant counterpart oligo 4-M (≥3 peptides with ≥2-fold changes), and 13 of them related to knockdown-induced loss of binding (S1 Table and Fig 1C). Besides YY1, the transcription factor AP-2C (TFAP2C) was found to bind specifically to the oligo 4 probe containing the YY1-B motif, but not to oligo 4-M (S1 Table). Another TFAP2 family member, TFAP2A, exhibited a higher binding affinity to oligo 4 than to the mutated oligo (Fig 1C).

Furthermore, the binding of TFAP2A and TFAP2C to the YY1-B motif was confirmed using nuclear extracts from two types of HPV-positive cell types (Figs 1D and S1B). Notably, TFAP2C was not detected in the oligo 4-M pull-down samples by mass spectrometry (Fig 1C), probably owing to the high sequence similarity between TFAP2A and TFAP2C, which led to homologous peptides being predominantly assigned to TFAP2A. Interaction between YY1 and TFAP2A has been reported [22,23] raising the possibility that TFAP2C binding to oligo 4 might be YY1-dependent. However, oligo 5, containing YY1-A motif, failed to recruit either TFAP2A or TFAP2C (Fig 1A and 1E). Moreover, co-immunoprecipitation in CaSki cells didn't reveal an interaction between YY1 and TFAP2A or TFAP2C. Binding of TFAP2 independent of YY1 and E7 was further confirmed by siRNA knockdown (Fig 1F). Taking together, these findings suggest that TFAP2 proteins bind to the core region of lnc-FANCI-2 promoter independent of the YY1 or E7.

### TFAP2 competes with YY1 in binding to YY1-B motif

TFAP2 proteins recognize specific DNA binding motifs. Accordingly, using JASPAR database, we identified a promising motif (score=384) at the 5′ end of the YY1-B region. Interestingly, this predicted TFAP2-binding motif overlaps with the YY1-B motif (Fig 2A) and is highly conserved among vertebrate species (Fig 2A). To test whether TFAP2 binding depends on this motif, we introduced two additional mutations within the TFAP2-binding motif (Fig 2A) and performed oligo pull-down assays. All three mutants displayed reduced or abolished binding to TFAP2A and TFAP2C. In contrast, mutations in oligo 4-M led a complete loss of YY1 binding, whereas oligo 4-M1 retained partial binding (Fig 2B). These results suggest that TFAP2 binding is sequence dependent and relies on a motif overlapping with YY1-B, with the shared "GGC" sequence being crucial for both TFAP2 and YY1 binding (Fig 2B).

To determine whether TFAP2 and YY1 bind to the overlapping site competitively (Fig 2C, left) or cooperatively (Fig 2C, right), we first assessed their relative binding affinities. By oligo titration in the pull-down assay (Fig 2D), we observed that the DNA-binding affinity of TFAP2A and TFAP2C to the oligo 4 was much lower than that of the YY1 protein. Furthermore, increasing levels of exogenous YY1 led to decreased TFAP2 binding in the pull-down assay (Fig 2E), indicating that TFAP2 and YY1 proteins compete to bind to the oligo 4 through an overlapping motif. Consistently, electrophoretic mobility shift assays (EMSA) further supported this competition, as exogenous YY1 could disrupt the potential TFAP2–oligo4 complex (S1C Fig). In addition, we performed oligo pull-down assays using nuclear extracts from control and YY1-depleted CaSki cells (S1D Fig). While no displacement was observed under high oligo conditions (5 μg or 1 μg), reducing the oligo amount to 0.2 μg or 0.04 μg resulted in significant TFAP2 enrichment upon YY1 depletion (lane 5 vs. lane 9),

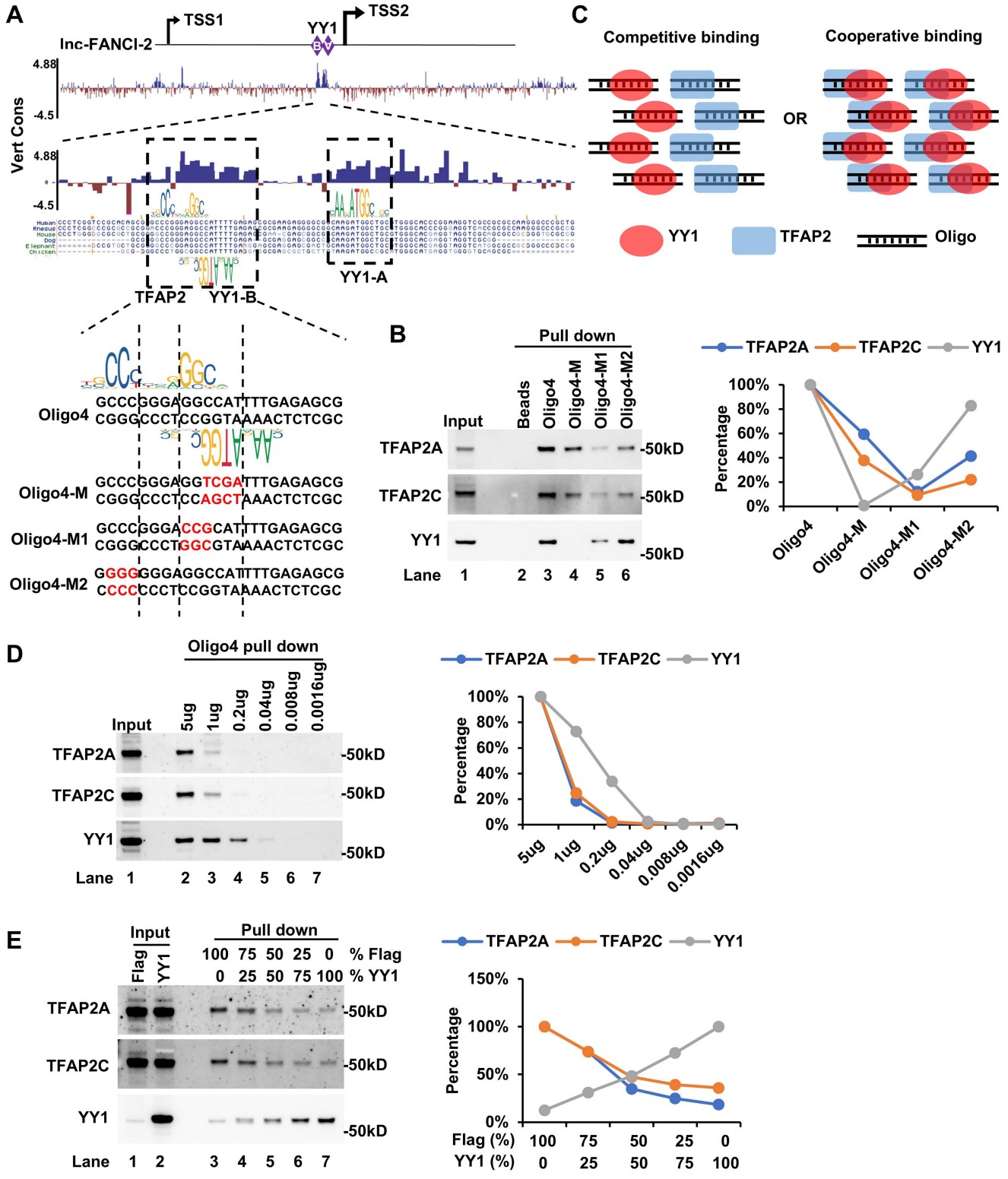

**Fig 2. Competitive binding of TFAP2 and YY1 to the YY1 motif B upstream of the lnc-FANCI-2 transcription start site 2 (TSS2).** (A) Diagrams of two YY1 binding motifs upstream of the TSS2 which are essential for the expression of lnc-FANCI-2 [16] and are highly conserved. Vert cons is "Conservation Track" adapted from UCSC Genome Browser, which shows multiple alignments of vertebrate species and measurements of evolutionary

conservations using phastCons and phyloP (panel top). Showing below the vert cons is a potential TFAP2 binding motif overlapping with the YY1 motif B and conservation in vertebrates along with biotinylated DNA oligos containing wild-type or mutation (M) YY1 or TFAP2 binding motifs used for DNA oligo pulldown assays. (B) TFAP2A, TFAP2C, and YY1 interaction with YY1-B motif by DNA oligo pulldown assays with fractionated CaSki nuclear extracts. The right panel shows the percentage of the indicated proteins. (C) Diagrams show hypothetic competitive binding and cooperative binding of YY1 and TFAP2 to the YY1-binding motif B. (D) YY1 has a higher binding affinity than TFAP2A and TFAP2C to the YY1-B. The YY1-B-bearing oligo 4 was used in the pulldown-Western blot assays. The right panel is the percentage of the indicated proteins in the pulldown-Western blot assays relative to that in the oligo4 at 5 ug. (E) YY1 competes with TFAP2 in binding to the YY1-B. Nuclear extracts from CaSki cells transfected with YY1-Flag (YY1) or Flag-control (Flag) vectors, respectively, were mixed in an indicated proportion followed by oligo4 pulldown-Western blot assays. The right panel shows the percentage of the indicated proteins in the competitive binding assays with the indicated nuclear fractions of Flag and YY1.

demonstrating the competitive relationship under appropriate thresholds (S1D Fig). Taken together, these results support a model in which TFAP2 and YY1 bind to oligo 4 in a mutually exclusive manner through a shared DNA motif.

### TFAP2 silences lnc-FANCI-2 expression in primary epithelial cells

Our competition model predicts that the outcome should depend on the relative levels of YY1 and TFAP2 in a cell. Since HPV infection promotes YY1 expression—leading to elevated lnc-FANCI-2 levels in cervical cancer cells [16]—we next sought to determine how TFAP2 binding and regulatory function vary between normal and HPV-transformed cellular contexts. To test this, we compared TFAP2 activity in human primary foreskin keratinocytes (HFKs) and CaSki cells. HFKs are primary epithelial cells that lack HPV infection but have been widely used as an in vitro model for HPV studies [24]; they exhibit relatively low expression of both YY1 and lnc-FANCI-2. CaSki cells are a cervical cancer cell line with integrated HPV16 DNA and active early viral gene expression, display high levels of YY1 and lnc-FANCI-2 [16]. Chromatin immunoprecipitation (ChIP) assays showed that YY1 binding to the lnc-FANCI-2 promoter was weaker in HFK cells than in CaSki cells, whereas TFAP2 bound to the lnc-FANCI-2 promoter at similar levels in both cell types (Fig 3A–3B). Another specific TFAP2 binding event was identified at a distinct TFAP2 motif within the promoter region (Fig 3C–3D, oligo1), which may explain the unchanged TFAP2 ChIP enrichment observed in CaSki cells. As previously shown, YY1 binding to YY1-B motif activates lnc-FANCI-2 expression in CaSki cells [16]. Therefore, this competitive binding between YY1 and TFAP2 to the same YY1-B motif might result in a cell-type-specific regulation of lnc-FANCI-2 expression. In fact, TFAP2 functioned as a transcriptional repressor of lnc-FANCI-2 in HFKs where YY1 level is low, but not in CaSki cells where YY1 level is high. This is supported by knockdown of TFAP2 increased lnc-FANCI-2 expression (Fig 3E) in HFKs, but had no significant effect in CaSki cells.

Accordingly, we proposed a steric competition model (Fig 4). In HPV-negative keratinocytes cells, where YY1 expression is low, TFAP2 functions as a repressor by binding to the lnc-FANCI-2 promoter (Fig 4, top). Upon hrHPV infection, the increased YY1-due to reduced miR-29a expression and enhanced recruitment by HPV E7 [16,25] (Fig 4, bottom)-outcompetes TFAP2 for binding at the promoter core region and transactivates the lnc-FANCI-2 promoter. This upregulated lnc-FANCI-2 expression may contribute to hrHPV-induced carcinogenesis by modulating the intrinsic IFN response and RAS signaling pathways [20].

### Genome-wide screening of YY1 and TFAP2 competitive binding sites

Building on our findings at the lnc-FANCI-2 promoter, we hypothesized that YY1 and TFAP2 competition represents a broader regulatory mechanism, with competition binding sites (CBS) potentially existing at additional genomic loci to alter gene expression programs during hrHPV-induced carcinogenesis. To test this genome-wide, we performed a bioinformatic screen using high-stringency JASPAR scores for YY1 (MA0095.3) and TFAP2 (MA0810.1) binding sites. This analysis identified 3726 potential CBSs genome-wide, with 49.0% of them are located in intergenic regions (Fig 5A and S2 Table). Functional enrichment analysis suggested that these CBSs are potentially involved in multiple carcinogenesis-related pathways, including cell junction assembly, cell-cell adhesion via plasma membrane adhesion molecules, and keratinocyte

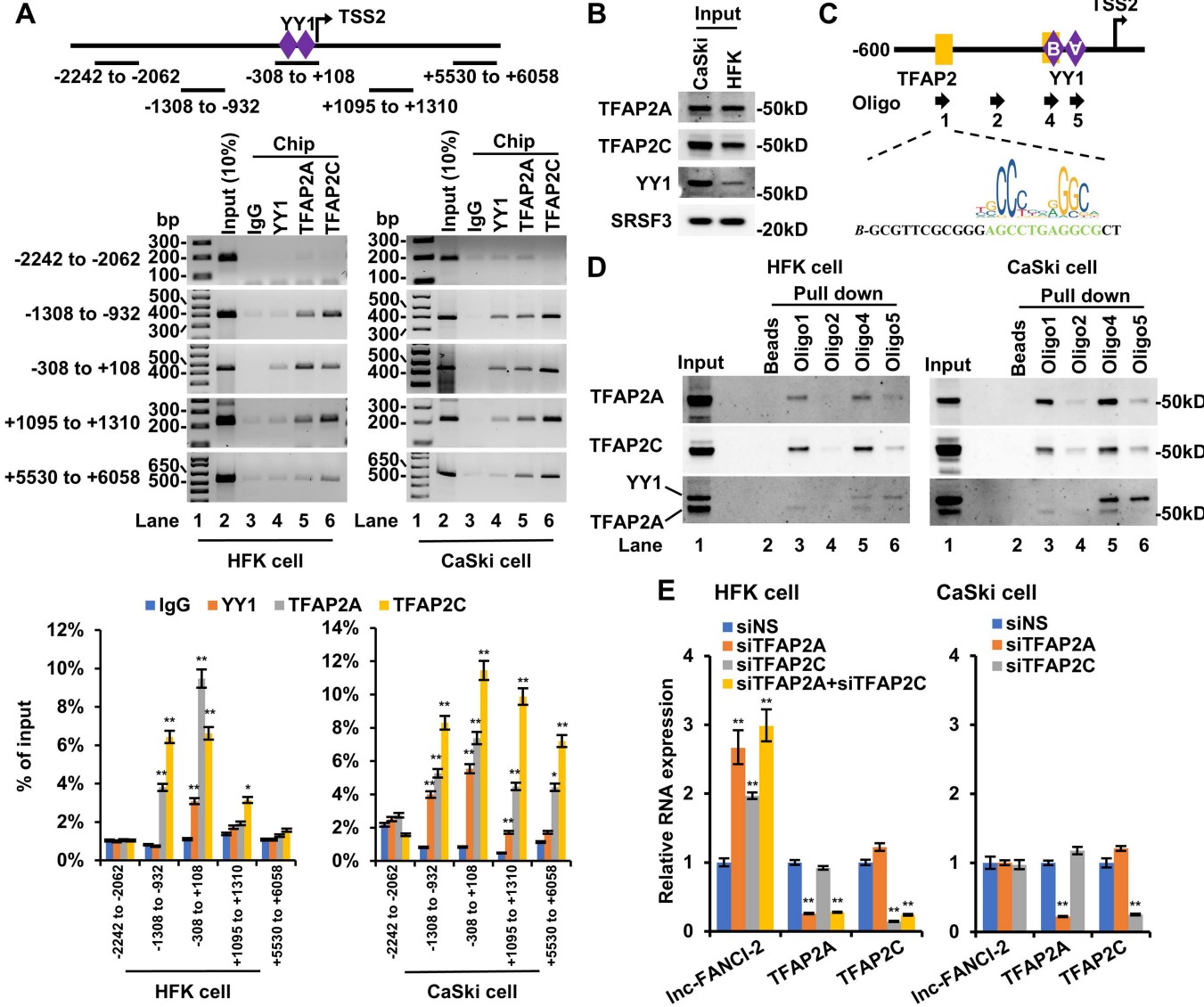

**Fig 3. TFAP2 binding and lnc-FANCI-2 expression in human primary foreskin keratinocytes (HFK) and HPV16+ CaSki cells.** (A) TFAP2A, TFAP2C, and YY1 are strongly associated with lnc-FANCI-2 promoter in both HFKs and CaSki cells. ChIP-PCR was performed with an anti-YY1, anti-TFAP2A, or anti-TFAP2C antibody on chromatin preps from CaSki cells or HFK cells by five pairs of the primers crossing the regions as diagramed. The lower panel of the bar graph shows the percentage of input DNA to each ChIPed target DNA region by individual antibodies over IgG control. (B-D) Identification of another TFAP2 binding motif further upstream of the lnc-FANCI-2 TSS2 using the nuclear extracts from HFK and CaSki cells. The nuclear input proteins were detected by Western blot, and SRSF3 serves as a loading control (B). Numbered arrows are DNA oligos corresponding to TFAP2 or YY1 binding motifs used for the pulldown assays (C) with a potential TFAP2 binding motif sequence in oligo 1 shown below. TFAP2 in HFK and CaSki nuclear extracts bound to oligos 1 and 4 shows in panel D. (E) TFAP2A and TFAP2C appear being repressive on lnc-FNACI-2 expression in HFK, but not in CaSki cells. The lnc-FANCI-2 expression in CaSki or HFK cells with or without KD of TFAP2A or TFAP2C were measured by RT-qPCR. KD efficiency of TFAP2A or TFAP2C was validated by RT-qPCR.

proliferation (Fig 5B and S3 Table), although further experimental validation is needed. Lnc-FANCI-2 has been associated with the expression of adhesion and membrane proteins in RAS signaling and epithelial-mesenchymal transition, indicating a complicated regulatory network behind these CBSs and their targets. To refine our list, we cross-referenced

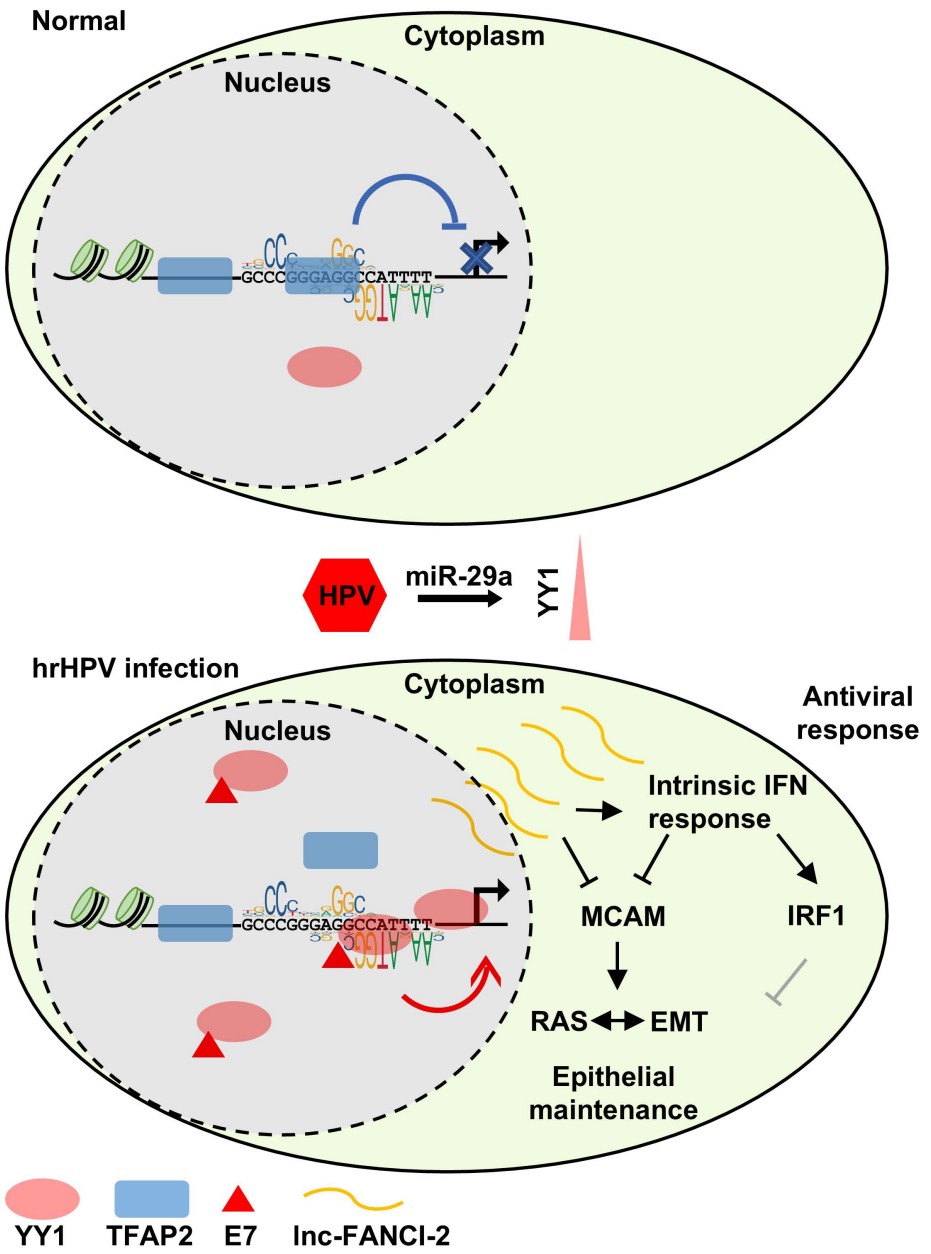

**Fig 4. A model about competitive binding of YY1 with TFAP2 in lnc-FANCI-2 expression in hrHPV infected cells.** IFN, interferon; EMT, Epithelial mesenchymal transition.

these predicted CBSs with the CHIP-seq peaks from the ENCODE Uniform TFBS composite track. 30 high-confidence CBSs were identified, (Fig 5A, Venn diagram, bottom and S4 Table), most of which were located in the promoter regions (Fig 5A). These CBSs frequently overlapped with both YY1 and TFAP2 CHIP peaks and exhibited epigenetic hallmarks of transcriptionally active regions, such as H3K27ac or H3K4Me3 (S2 Fig). From these, 20 CBSs were selected for further validation, and we found that all CBS-derived oligos were confirmed as YY1-binding sites (Fig 5C). Six CBSs, located in the promoter regions of LRRC37A3, LYRM4, CNOT1, PPP1R15B, AKAP8L, and VPS11, also showed TFAP2 binding

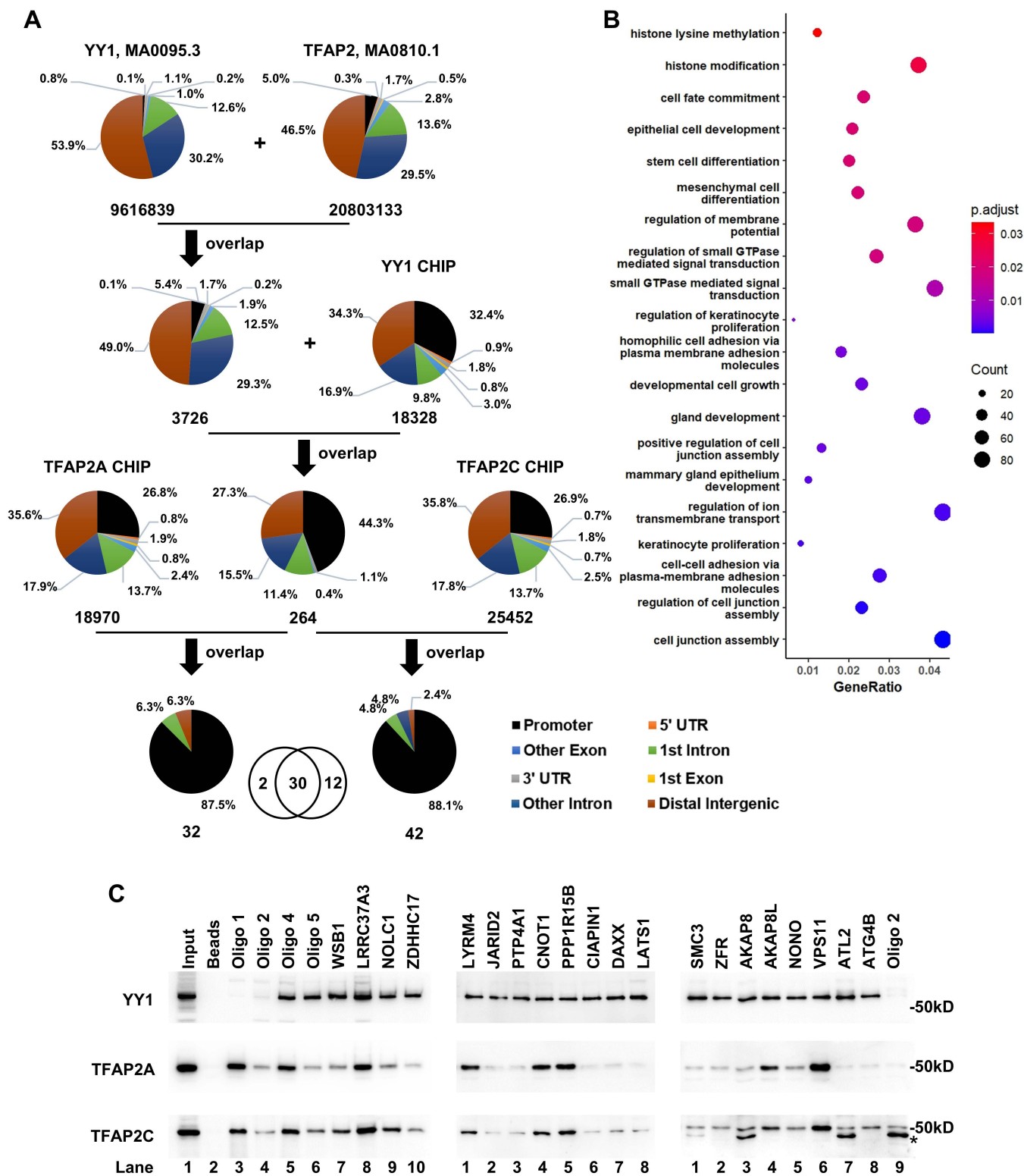

**Fig 5. Identification of genome-wide competitive YY1 and TFAP2 binding sites.** (A) A workflow for genome-wide screening of YY1 and TFAP2 CBS. Overlapping YY1 (MA0095.3, score > 520) and TFAP2 (MA0810.1, score > 250) motifs by JASPAR was intersected with CHIP peaks of YY1, TFAP2A and TFAP2C from ENCODE Uniform TFBS composite track. The proportion of annotated binding motifs was analyzed by pie graphs, and the number below

each pie shows the total motif number. Venn diagram at the bottom represents 30 CBS identified by overlapping of YY1 with both TFAP2A and TFAP2C CHIP peaks. (B) Functional GO enrichment analysis of 3726 potential CBS of YY1 and TFAP2 by JASPAR (S3 Table). Bubble plot shows 20 pathways selected from the top list. (C) Validation of randomly selected 20 of 30 CBS identified in the Venn diagram (A) by DNA oligo pulldown and Western blot assays. * Undetermined protein.

and overlapping YY1 and TFAP2 CHIP peaks, and H3K27ac or H3K4Me3 marks in the promoter regions (Figs 6A–6B and S2, top panel). The overlapped YY1 and TFAP2 binding motifs are illustrated on the JASPAR 2020 TFBS track (Figs 6A–6B and S2, bottom panel). Subsequently, we selected two CBSs, one from PPP1R15B and the other from LRRC37A3 promoter, for further analysis. Pull-down assays showed that TFAP2 and YY1 bind competitively to each CBS-containing DNA oligo (Fig 6C–6D). Interestingly, silencing YY1 using two independent siRNAs altered the expressions of both PPP1R15B and LRRC37A (Fig 6E–6F). High expression levels of either PPP1R15B or LRRC37A were correlated with poor survival outcomes in patients with cervical cancer (Fig 6G–6H).

## Discussion

The expression of lnc-FANCI-2 is associated with the progression of cervical lesions caused by hrHPV infection [16]. Acting as a host defense lncRNA, lnc-FANCI-2 preserves epithelial integrity by suppressing RAS signaling and boosting intrinsic interferon responses, thereby counteracting hrHPV-induced oncogenic stress [20]. However, lnc-FANCI-2 induction is largely independent of the tumor suppressors pRB and p53, suggesting other host factors play a role in the maintenance epithelial homeostasis during HPV-mediated cervical carcinogenesis. Given that YY1 binding to the promoter core region is crucial for lnc-FANCI-2 transactivation, we sought to identify additional cofactors involved in this regulatory mechanism. Rather than uncovering transactivation partners, we found two TFAP2 family members binding to an overlapping YY1 motif and function as transcriptional repressors in primary epithelial cells.

TFAP2 proteins share a conserved DNA-binding motif and are capable of forming homo- or hetero-dimeric complexes to regulate target gene expression [26,27]. As pioneer transcriptional factors, they facilitate chromatin accessibility for other regulators [28–30], suggesting that TFAP2 may occupy the lnc-FANCI-2 promoter in normal keratinocytes to maintain a permissive chromatin state. The transcriptional outcome of lnc-FANCI-2 depends on the co-activators or co-repressors recruited by TFAP2 [27]. Functional binding sites for TFAP2A or TFAP2C in the promoters of keratinocyte-specific genes are crucial for epidermal maturation [29,31–33]. We consider lnc-FANCI-2 a keratinocyte-specific gene, as its promoter is selectively active in keratinocytes (S3 Fig). Thus, in normal keratinocytes TFAP2 likely maintains repression of the lnc-FANCI-2 transcription. Under these conditions, YY1 mRNA is suppressed by high levels of miR29a [16]. Upon HPV infection, the upregulated YY1 protein competes with TFAP2 for occupancy of the YY1-B motif. While TFAP2 remains bound to other sites in the lnc-FANCI-2 promoter (Fig 3C and 3D), this residual TFAP2 binding appears insufficient to repress transcriptional activity.

Although steric competition between transcription factors has been proposed, it applies between factors with similar DNA-binding specificities, such as MYC versus MiT/TFE or KLF1 versus KLF3 [34,35]. Spatial restrictions on transcription factors can occur if binding sites are sufficiently close, though little is known about competition at overlapping motifs. Here, we developed a DNA oligo pull-down assay coupled with mass spectrometry and antibody-based detection to evaluate mutual exclusivity binding at shared sites (Fig 2). This approach circumvents the need for protein purification and allows direct comparison of binding affinities. We extended our analysis for additional CBSs across the genome, focusing on sites with at least one nucleotide overlap for simplicity. However, many potential CBSs might have been overlooked due to unknown spatial constraints for mutually exclusive binding. Among the validated sites, a few CBSs in the promoter regions were confirmed further. However, it is likely that CBSs in distal enhancers or insulators also contribute to chromatin conformation, particularly given YY1 is a known chromatin structural regulator [36]. Additionally, gene ontology analysis indicated that CBSs may be involved in diverse biological processes beyond cancer, including axon development and synaptic organization (S3 Table), consistent with the broad expression patterns of both YY1 and TFAP2.

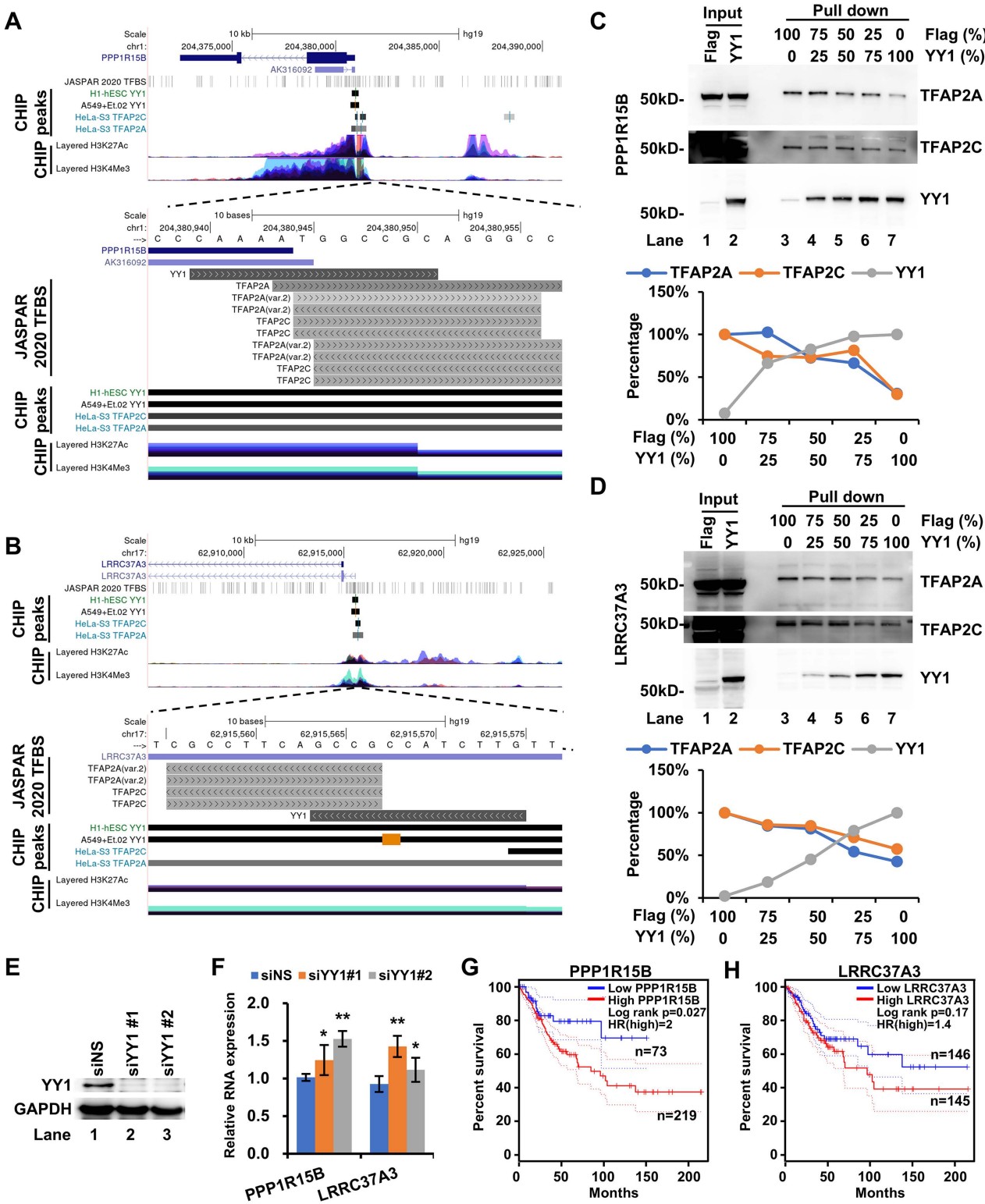

**Fig 6. Characterization of two CBS in the PPP1R15B or LRRC37A3 promoter for competitive binding of TFAP2 and YY1.** (A-B) CBS in the PPP1R15B (A) or LRRC37A3 (B) promoter illustrated with UCSC Genome Browser regulation tracks of JASPAR 2020 TFBS, CHIP peaks from indicated cells and H3K4Me3 or H3K27Ac marks from ENCODE's 7 cell lines. The upper panel views in each panel show the genome locus of each gene

(GRCh37/hg19). The lower panel views in each panel are zoomed in the YY1 and TFAP2 motifs showing all regulation tracks. (C-D) Competition of YY1 and TFAP2 binding to the CBS identified in the PPP1R15B (C) or LRRC37A3 (D) promoter by DNA oligo pulldown-Western blot assays. Nuclear extracts of HeLa cells transfected with a YY1-Flag or an empty Flag-control vector were mixed in the indicated ratio for the indicated DNA oligo pulldown and Western blot assays for overexpressed YY1-Flag and endogenous TFAP2A and TFAP2C. The lower panel shows the percentage of the indicated proteins in the competitive binding assays. See individual oligo sequences in S5 Table. (E-F) PPP1R15B or LRRC37A3 were significantly upregulated in YY1 deficiency cells. The YY1 knockdown efficiency in HeLa cells was verified by Western blot (E). The expressions of PPP1R15B or LRRC37A3 in YY1 deficiency cells were quantified by RT-qPCR. (G-H) Survival analysis of PPP1R15B (G) or LRRC37A3 (H) in 292 cervical squamous cell carcinoma cases from the cancer genome atlas (TCGA) datasets by GEPIA web server (http://gepia.cancer-pku.cn/about.html).

While our study elucidates a mechanistic model of transcriptional competition, the direct functional impact of the YY1/ TFAP2 axis on HPV-induced tumorigenesis warrants further investigation. Future studies employing more complex biological systems will be essential to validate the oncogenic role of this pathway. We propose to utilize HPV-positive organoid models and patient-derived xenografts to functionally test whether the competitive displacement of TFAP2 by YY1 drives hallmark cancer phenotypes such as sustained proliferation, invasion, and in vivo tumor growth. In summary, our study uncovers a novel mechanism of transcriptional antagonism between YY1 and TFAP2 in hrHPV-induced carcinogenesis. This antagonism governs the expression of lnc-FANCI-2 and other genes relevant to tumor progression. Our findings suggest that targeting YY1 and TFAP2 in hrHPV-infected cells may offer new strategies for therapeutic intervention.

## Materials and methods

### Cell cultures

CaSki cells and HeLa cells (ATCC) were maintained in Dulbecco's modified Eagle's medium (DMEM) (Thermo Fisher Scientific) supplemented with 10% fetal bovine serum (FBS) and 1% penicillin-streptomycin, and cultured at 37 °C with 5% $CO_2$.

Human primary foreskin keratinocytes (HFKs) were cultured on mitomycin C-treated 3T3 cells (J2 strain) in mixed F medium: DMEM (3:1) supplemented with of 5% FBS, 0.4 µg/mL hydrocortisone, 5 µg/mL insulin, 8.4 ng/mL cholera toxin, 10 ng/mL epidermal growth factor (Invitrogen), 24 µg/mL adenine, and 5 µmol/L Rho kinase inhibitor, Y-27632 (#ALX-270–333; Enzo Life Sciences) at 37 °C with 5% $CO_2$ [37].

### Biotin-labeled double-stranded DNA oligo pull down

DNA oligonucleotide pull-down assays were performed as described previously [16]. Cell extracts were isolated from CaSki or HFK cells using the Nuclei ez prep kit (NUC-101; Sigma) according to the manufacturer's protocol. The nuclear extract was resuspended in cold RIPA buffer (BP-115X), centrifuged at 12000 × g for 5 min, and transferred to new tubes. The insoluble nuclear pellets were resuspended in the same amount of RIPA buffer. Cytoplasmic, soluble, and insoluble nuclear fractions were verified by immunoblot analysis to confirm the fractionation efficiency. The protein concentration was determined using the Micro BCA Protein Assay (#23235; PIERCE). The soluble nuclear extract was then used to pull down the DNA oligo. To anneal double-stranded oligos, 200 µM (~1.5 µg/µLl) of biotinylated sense oligo and anti-sense oligo were incubated at 95 °C for 10 min and cooled down to room temperature. Pierce NeutrAvidin Agarose (200 µl; 50% slurry; #29201) was washed three times and resuspended in 1 mL wash buffer (20 mM Tris-HCl [pH 7.5], 100 mM NaCl, 1 mM MgCl2, 0.5 mM EDTA, 0.5 mM DTT). Then, the beads were coated with double-stranded oligos by incubating with 10 µg of double stranded oligos (unless stated otherwise) for 1–2 h at room temperature. Oligo-coated beads were washed with buffer three times and then mixed with 200 ug of nuclear extract in 1 mL of pull-down buffer (20 mM Tris-HCl [pH 7.5], 100 mM NaCl, 1 mM MgCl2, 0.5 mM EDTA, 0.5 mM DTT, 4% glycerol, 10 µg/mL Poly dI-dC, 1 × Roche's protease inhibitor cocktail) and incubated overnight at 4 °C. The DNA oligo-associated complexes on the beads were washed six times and divided into two parts. One part of the beads was dissolved in 50 µL of 2 × SDS protein sample buffer and subjected to SDS-PAGE and western blot. The remaining beads were used for mass spectrometric analysis as described below.

## DNA oligo pull down coupled mass spectrometry

The beads from the DNA oligo pull-down assay described above were resuspended in buffer (25 mM $NH_4HCO_3$, pH 8.4) and heated at 95 °C for 5 min to denature the proteins. The samples were digested overnight with 2 µg of trypsin at 37 °C. The supernatant containing the tryptic digest was obtained after the centrifugation of the beads. The beads were washed twice with buffer (25 mM $NH_4HCO_3$, pH 8.4) to obtain maximum yield. The tryptic digest was desalted using C18 columns (Thermo Fisher Scientific) and lyophilized. Mass spectrometry (MS) acquisition and data analysis were performed as follows. The dried samples were reconstituted in 0.1% trifluoroacetic acid (TFA) and subjected to nanoflow liquid chromatography (Thermo Easy nLC 1000; Thermo Fisher Scientific) coupled with high-resolution tandem MS (FUSION; Thermo Fisher Scientific). MS scans were performed in an Orbitrap analyzer at a resolution of 60 000 with an ion accumulation target set at $4e^5$ over a mass range of 375–1500 m/z, followed by MS/MS analysis in an ion trap with an ion accumulation target set at $1e^4$. The MS2 precursor isolation width was set up at 1.6 m/z, the normalized collision energy was 30, and charge state 1 and unassigned charge states were excluded. The acquired MS/MS spectra were searched against a human UniProt protein database using SEQUEST, and the resulting peptides were filtered at a maximum of 1% FDR using the percolator validator algorithms in the Proteome Discoverer 2.2 software (Thermo Fisher Scientific). The precursor ion tolerance was set at 10 ppm, fragment ion tolerance was set at 0.6 Da along methionine oxidation was included as a dynamic modification.

## Transfection

Four siRNA pools, siYY1, siTFAP2A, siTFAP2C, and nonspecific siRNA (Figs 1 or 3), were predesigned siRNAs (siGE-NOME SMARTpool; Dharmacon). The other siRNAs, siYY1#1 and siYY1#2, used in Fig 6, were designed and synthesized by Sangon Biotech Co., Ltd. (Shanghai, China), and the sequences are listed in S5 Table. The siRNAs described above were transfected into CaSki, HFK, or HeLa cells using LipoJet in vitro transfection reagent (SignaGen, MD, USA). YY1-Flag [16] or Flag-control cells were transfected into CaSki or HeLa cells using the Lipofectamine 2000 Transfection Reagent (#11668027; Thermo Fisher Scientific).

## RT-qPCR

The detection of lnc-FANCI-2 by RT-qPCR was performed as described previously [16]. Primers designed for lnc-FANCI-2 are listed in S5 Table. Briefly, 2 µg total RNA was converted to cDNA using Superscript First-stand Synthesis kit (#11904018; Thermo Fisher Scientific). qPCR was performed using TaqMan Gene Expression Master Mix (#4369016; Applied Biosystems, Waltham, MA, USA) on a StepOne Plus Real-Time PCR system (Applied Biosystems). TFAP2A, TFAP2C, and GAPDH were pre-designed using TaqMan Gene Expression Assays (#4331182, Applied Biosystems). Fold changes were calculated with the control group as the reference using the 2-ΔΔCt method. The data were normalized first to the values for GAPDH and then to the median value for control samples. Data are plotted as a bar graph with the mean±SD for each group.

The expression of PPP1R15B and LRRC37A was quantified using a SYBR Green qPCR assay. Briefly, 2 µg total RNA was converted to cDNA using HiScript II Q RT SuperMix for qPCR (#R222-01) from Vazyme Biotech Co., Ltd. (Nanjing, China). qPCR was performed using Hieff qPCR SYBR Green Master Mix (#11201ES03) from Yeasen Biotechnology Co., Ltd. (Shanghai, China) with qualified qPCR primers. The amplification efficiency of the qPCR primers was between 90% and 110% based on the slope of the standard curve, and no dimers or nonspecific bands were detected in the melting curve. Primers used are listed in S5 Table. Gene expression levels were calculated as previously described.

## Antibodies and immunoblot

Anti-YY1 (#66281–1-Ig), anti-TFAP2C (#14572–1-AP), and anti-p53 (#10442–1-AP) antibodies were purchased from Proteintech (Rosemont, IL, USA). Anti-tubulin (#T5201) antibody was purchased from Sigma-Aldrich (St. Louis, MO, USA).

Anti-TFAP2A (AP-2α/β, #sc-70361) was from Santa Cruz (Dallas, TX). Anti-TFAP2A (AP-2α, #3215), anti-CCND2 (Cyclin D2, #3741), anti-Akt (#9272), and anti-phospho-Akt (#9271) antibodies were purchased from Cell Signaling (Danvers, MA, USA). Anti-SRSF3 (#NBP2–76892) antibody was purchased from NOVUS (Littleton, CO, USA). Immunoblot analysis was performed for individual proteins as previously described.

### Chromatin immunoprecipitation (ChIP)

Chromatin immunoprecipitation was performed as described previously using the ChIP-IT High Sensitivity (HS) Kit (Active Motif, Carlsbad, CA, # 53040) [16]. CaSki or HFK cells ($2 \times 10^7$) in T175 flasks were fixed with 2.5 mL of complete cell fixation solution for 15 min, followed by the addition of 1 mL of stop solution for 5 min at room temperature. Fixed cells were scraped, washed, and resuspended in 10 mL of chromatin prep buffer containing a protease inhibitor cocktail (PIC) and PMSF on ice. After 10 mins incubation, cell nuclei were isolated by 15 times passage of cells through a 25-gauge needle, and then resuspended in 500 μL of ChIP buffer in presence of 5 μL of PIC and 5 μL of 100 mM PMSF. Nuclear extracts were sonicated using 30 strokes at Level 4 on a Sonic Dismembrator (Model 100; Thermo Fisher Scientific). Input DNA was generated from 25 μL of chromatin preparation and shearing efficiency was examined by DNA electrophoresis. To perform CHIP, 50 μL of chromatin preparation was incubated with 4 μg of anti-YY1 (#66281–1-Ig; ProteinTech), anti-TFAP2C (#14572–1-AP; ProteinTech), or anti-TFAP2A (#3215; Cell Signaling Technology) antibodies overnight at 4 °C, then mixed with 30 μL of protein G agarose beads at 4 °C for 3 h. The ChIP-treated DNA was then eluted in elution buffer AM4, and DNA cross-links were reversed by proteinase K treatment and purified by column clean-up. PCR amplification was performed to determine the amount of ChIPed DNA using primers spanning the distal promoter. Primers used are listed in S5 Table. ChIP enrichment was calculated using the PCR band intensity after normalization to the input band intensity.

### Genome-wide screening of YY1 and TFAP2 CBS

The genome-wide predicted motifs of YY1(MA0095.3) and TFAP2 (MA0810.1) were downloaded from the JASPAR Genome Browser tracks (http://expdata.cmmt.ubc.ca/JASPAR/downloads/UCSC_tracks/2022/hg19/) and annotated using the ChIPseeker R/Bioconductor package. The YY1 and TFAP2 CBSs were identified by applying two criteria: 1) at least one nucleotide of the YY1 and TFAP2 motifs overlapped, and 2) the motif JASPAR score of YY1 was greater than 520 and that of TFAP2 was greater than 250. The functional GO enrichment analysis was performed using ClusterProfiler. The CHIP-seq peaks of YY1 (H1-hESC), TFAP2A (HeLa-S3), and TFAP2C (HeLa-S3) were downloaded from the ENCODE Uniform TFBS composite track (http://hgdownload.soe.ucsc.edu/goldenPath/hg19/encodeDCC/wgEncodeAwgTfbsUniform/) and annotated using the ChIPseeker R/Bioconductor package. The CBSs were identified using JASPAR overlapped with the annotated CHIP peaks.

### Supporting information

**S1 Table. Identified proteins in DNA oligo pulldown-coupled mass spectrometry.**
(XLSX)

**S2 Table. 3726 potential YY1 and TFAP2 competition sites by JASPAR.**
(XLSX)

**S3 Table. Gene Ontology (GO) analysis of 3726 TFAP2 and YY1 competition sites.**
(XLSX)

**S4 Table. High-confidence competition sites overlapping with CHIP peaks.**
(XLSX)

**S5 Table. Oligo information.**
(DOCX)

**S1 Fig. (A) E7 siRNA targets both E6 and E7 transcripts.** Soluble nuclear extracts were prepared from E7-siRNA transfected CaSki cells, and the expression levels of HPV16 E6, E7, and p53 were evaluated by western blot using anti-E6 (#GTX132686, GeneTex), anti-E7 (#GTX637546, GeneTex), and anti-p53 (#10442–1-AP, Proteintech) antibodies. hnRNPC1/C2 (#68447–1-Ig, proteintech) was used as an internal control. (B) **Binding of TFAP2A and TFAP2C to the YY1 motif B.** Pulldowns were performed using oligo4, oligo4-M, oligo5, and oligo5-M with fractionated nuclear extracts isolated from HeLa cells with the indicated siRNA knockdown. (C) **Validation of TFAP2 and YY1 competition for oligo4 by EMSA.** FAM-labeled oligo4 was synthesized by Sangon Biotech, and unlabeled oligo4 was used as a competitor. Nuclear extracts containing YY1-Flag (YY1) or Flag-control (Flag) were prepared as described in Fig 2E. Binding reactions were performed by incubating 0.1 pmol of FAM-labeled oligo4 with the indicated nuclear extracts in 1 × EMSA/Gel-Shift binding buffer (#GS005, Beyotime Biotech) for 20 minutes at room temperature. Samples were then resolved on a 4–20% BeyoGel TBE PAGE gel (#D0185S, Beyotime Biotech). The gel was scanned using an Amersham Typhoon imager (GE Healthcare). Lane 1 contained 10 pmol of unlabeled oligo4 as a competitor. NE: nuclear extract. (D) **Validation of TFAP2 and YY1 competition via oligonucleotide titration assay.** Soluble nuclear extracts were prepared from CaSki cells treated with non-specific siRNA (siNS) or YY1 siRNA (siYY1). LE, long exposure; SE, short exposure.
(TIF)

**S2 Fig. UCSC Genome Browser regulation tracks of JASPAR 2020 TFBS, CHIP peaks and H3K4Me3 or H3K27Ac marks from ENCODE's 7 cell line datesets.** The upper panel views in each panel show the genome locus of each gene (GRCh37/hg19) and the lower panel views in each panel are zoomed in the YY1 and TFAP2 binding motifs with all regulation tracks.
(TIF)

**S3 Fig. Chromatin State Segmentation by HMM from ENCODE/Broad displays a chromatin state segmentation for each of nine human cell types.**
(TIF)

## Acknowledgments

We gratefully acknowledge the significant contributions to this work from Dr. Zhi-Ming Zheng's laboratory (Tumor Virus RNA Biology Section, HIV Dynamics and Replication Program, CCR/NCI/NIH).

## Author contributions

**Conceptualization:** Yi Liu, Haibin Liu.

**Data curation:** Haibin Liu.

**Formal analysis:** Yi Liu, Shuang Ding.

**Funding acquisition:** Haibin Liu.

**Investigation:** Yi Liu, Shuang Ding, Haibin Liu.

**Methodology:** Yi Liu, Haibin Liu.

**Validation:** Yi Liu, Haibin Liu.

**Visualization:** Yi Liu, Haibin Liu.

**Writing – original draft:** Haibin Liu.

**Writing – review & editing:** Yi Liu, Shuang Ding, Haibin Liu.

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
