## [Decision Letter · Decision Letter 0]

6 May 2025

A genome-wide analysis of YY1 and TFAP2 competition on overlapping motifs reveals their roles in HPV-induced carcinogenesis

PLOS Pathogens

Dear Dr. Liu,

Thank you for submitting your manuscript to PLOS Pathogens. After careful consideration, we feel that it has merit but does not fully meet PLOS Pathogens's publication criteria as it currently stands. Therefore, we invite you to submit a revised version of the manuscript that addresses all of the points raised during the review process.

Please submit your revised manuscript within 60 days Jul 05 2025 11:59PM. If you will need more time than this to complete your revisions, please reply to this message or contact the journal office at plospathogens@plos.org. Please include the following items when submitting your revised manuscript:

We look forward to receiving your revised manuscript.

Kind regards,

Cary A. Moody

Academic Editor

PLOS Pathogens

Blossom Damania

Section Editor

PLOS Pathogens

Editor-in-Chief

PLOS Pathogens

orcid.org/0000-0003-2946-9497

Editor-in-Chief

PLOS Pathogens

orcid.org/0000-0002-7699-2064

**Journal Requirements:**

1) We noticed that you used the phrase 'data not shown' in the manuscript. We do not allow these references, as the PLOS data access policy requires that all data be either published with the manuscript or made available in a publicly accessible database. Please amend the supplementary material to include the referenced data or remove the references.

2) Please ensure that your paper does not exceed the word limit of 1,800 words for this article type. 

Potential Copyright Issues:

i) Figures 2, and S1A. Please confirm whether you drew the images / clip-art within the figure panels by hand. If you did not draw the images, please provide (a) a link to the source of the images or icons and their license / terms of use; or (b) written permission from the copyright holder to publish the images or icons under our CC BY 4.0 license. Alternatively, you may replace the images with open source alternatives. See these open source resources you may use to replace images / clip-art:

**Comments to the Authors:**

**Please note that one of the reviews is uploaded as an attachment.**

**Reviewers' Comments:**

Reviewer's Responses to Questions

**Part I - Summary**

Reviewer #1: In general, this is an interesting study in which they discovering new TFAP2 family members that compete with YY1 for binding at overlapping sites of the lnc-FANCI-2 promoter. Basically, in primary epithelial cells, TFAP2 binding led to lnc-FANCI-2 silencing. In contrast, in HPV-positive cancer cells, increased YY1 levels displaced TFAP2, alleviating repression. They also did a genome-wide predictions and identified thousands of YY1 and TFAP2 competition binding sites (CBSs), many overlapping with CHIP peaks for YY1, TFAP2A, and TFAP2C, predominantly in promoter regions. Even though these interesting findings, there are a few questions about presented data need to be addressed.

Reviewer #2: Comments on "A genome-wide analysis of YY1 and TFAP2 competition on overlapping motifs reveals their roles in HPV-induced carcinogenesis" by Liu et al.

This report investigates the regulatory mechanism of the long non-coding RNA lnc-FANCI-2 in HPV-driven cervical carcinogenesis. The report provides novel insights into the antagonistic roles of YY1 and TFAP2 in regulating lnc-FANCI-2 and broader transcriptional networks, supported by omics validation. The development of a DNA oligo pulldown method to map competitive TF binding is technically innovative and broadly applicable.

Overall, this manuscript presents valuable findings about the transcriptional regulation of lnc-FANCI-2 in the context of HPV infection and proposes a new model of transcription factor competition. However, the manuscript would benefit from clearer logic and more precise language to improve readability and understanding.

Reviewer #3: This is a follow-on study of the YY1-dependent transcriptional regulation of lnc-FANCI-2 transcription in HPV-driven cervical cancer. The authors identify several novel interacting proteins of the YY1 binding region of the lnc-FANCI-2 promoter using in vitro oligo pull down assays. The findings are interesting. However, they suggest that while HPVE7 is required for the binding of YY1 to the lnc-FANCI-2 promoter, that a competitive binder to the same region, TFAP2, is not affected by E7 expression levels. The data underpinning these findings are a little weak (and mostly buried in the supplemental figures). I do feel that for publication in PLoS Pathogens, the viral, host-pathogen interaction aspects of this study should be enhanced before it can be considered.

Some specific comments are listed below:

The YY1 binding sites in the lnc-FANCI-2 promoter need to be better introduced in the text – the results start with a statement about YY1-B – this is not easy to follow without an initial definition of YY1-A/B and their function.

Of the 28 proteins identified as specifically associated with the YY1-B motif, 12 of these proteins with ‘related to E7 KD-induced loss of binding’. It isn’t clear what these proteins are and what criteria have been used to assess dependence on E7 for binding; there are 13 proteins listed in table S1 that appear to be lost when E7 is depleted.

A western blot showing the efficiency of E6 and E7 depletion with siRNA is required. Throughout the manuscript, the siRNA is described as E7-specific but it could affect both E6 and E7. The only readout for the efficacy for this siRNA is nuclear p53 (fig S1B) but this is not a readout of E7 levels. It is misleading that the authors state that the binding of particular proteins to the oligo is E7-specific when E6 and/or E7 could be involved.

FigS1E appears to show that knockdown of E7 enhances YY1 binding to oligo 5. This is at odds with the authors’ previous manuscript showing that knockdown of E7 reduces YY1 binding to the lnc-FANC-2 promoter region (at YY1 sites A and B). Please can the authors explain.

What does a motif score of 384 mean? Without some context of the scale used for these scores, and how this number is derived, it is meaningless.

Fig 1B – the ‘reduced or lost’ binding of TRAF2C to the mutant oligos needs to be quanitifed. An experiment similar to those shown in Fig1D, comparing the WT and mutant oligos, would be helpful for this purpose.

Figure S2B needs a loading control.

The ChIP results in Fig S2A should be shown as bar charts – the data points are not contiguous and therefore a fitted line graph is not appropriate. The % of input in these graphs is add – reaching nearly 120% for TFAP2C in CaSki cells for example. ChIPs are notoriously inefficient and IP’ing 100% of input is very unlikely.

What are the sequences of the, presumably control, oligos 1 and 2 used in the in vitro pull down experiment in Fig 3C? How were these chosen? What cells were used for the nuclear extracts used in these experiments?

The ‘competitive binding’ shown in Figs 4C and D need to be quantified.

How was significance calculated in Fig4F – the differences look marginal.

**Part II – Major Issues: Key Experiments Required for Acceptance**

Reviewer #1: 1. In Figure S1D and S1E, better to provide TFAP2A data as well, making them consistent with other data. They also need to repeat these experiments (S1D and S1E) in other HPV+ cervical cancer cell lines such as HeLa to make sure: 1) whether it is cell line specific? 2) whether it is HPV 16 specific?

2. In Figure 3C, they described “all 20 CBS-derived oligos tested were YY1 binders”, what about TFAP2A and TFAP2C? It seems that they were also TFAP2A and TFAP2C binders?? Also, did they find some CBS which are only TFAP2A and TFAP2C dependent but not YY1 (as they found in the promoter of lnc-FANCI-2).

3. In Figure 4, based on results from 4F it seems that YY1 is a repressor for these targeted genes? (which is different from its role in lnc-FANCI-2). Also, it is confusing that “high level of YY1 in HPV+ cancer cells” while “high levels of either PPP1R15B or LRRC37A correlated with a worse survival rate in patients with cervical cancer”. Any contradiction?

Reviewer #2: 1. In the Introduction, the authors describe in separate paragraphs that the E6 and E7 genes are crucial for maintaining the transformed state in HPV-induced cancer cells, and that the YY1 gene is upregulated in HPV-positive cancer cells, activating cancer-associated lnc-FANCI-2. However, the lack of a clear connection between the discussion of E6 and E7 in the first paragraph and the upregulation of YY1 in the second paragraph makes it difficult to understand the background and significance of the research. It is suggested to include a discussion on how E7 regulates YY1, leading to the research objective of this study, in order to improve the logical flow of the writing.

Figure S1 and Figure 1 aim to identify the regulatory or binding proteins of the lnc-FANCI-2 core promoter. However, it is unclear why E7 knockdown was included in the DNA oligo pulldown-coupled mass spectrometry experiment. The knockdown of E7 does not appear to be directly related to the experimental objective described in this section. The authors should provide a clearer explanation of the experimental rationale behind the use of E7 knockdown.

2. The conclusion that TFAP2 competes with YY1 for binding to the YY1-B motif seems unconvincing. For example, in Fig. S2D, YY1 enrichment in CaSki cells (Oligo4) is significantly higher compared to HFK cells, but TFAP2A enrichment does not decrease; rather, it increases. Moreover, in Fig. S2E, when TFAP2A is knocked down in CaSki cells, one would expect YY1 binding to increase, and lnc-FANCI-2 expression to rise. In Fig. S2A-B, TFAP2 bound to the lnc-FANCI-2 promoter similarly in HFKs and CaSki cells, which raises doubts about the competition model. Therefore, the conclusion that TFAP2 competes with YY1 for binding requires additional experimental evidence.

3. The authors developed a DNA oligo pull-down method to compare binding affinities or determine mutually exclusive binding (Fig. 1). However, the reliability of this new method is not further evaluated. It is recommended to include a positive control group or incorporate additional methods for comparative analysis to strengthen the validity of the findings.

Reviewer #3: See above

**Part III – Minor Issues: Editorial and Data Presentation Modifications**

Reviewer #1: N/A

Reviewer #2: 1. The writing needs to be improved to enhance the readability and logical flow of the manuscript. For example, the first two sentences of the Results section, 'YY1-B is more critical for the lnc-FANCI-2 promoter activity. To further identify the regulatory or binding proteins of the lnc-FANCI-2 core promoter, we performed DNA oligo pulldown-coupled mass spectrometry using soluble nuclear extract (Nuc) from HPV16+ CaSki cells with or without E7 knockdown,' are not clearly connected. The statement about YY1-B does not logically lead to the description of the experiment in the second sentence.

2. The insertion of a working model in Figure 2 is not ideal for the structure of the manuscript. It would be better to introduce the relationships between YY1, TFAP2, E7, lnc-FANCI-2, and miR-29a in the Introduction and use Fig. 2 as an abstract diagram for clarity.

3. In several Western blot experiments, p53 expression is measured (e.g., Fig. S1B, where E7 siRNA targeting both E6 and E7 transcripts resulted in the accumulation of nuclear p53). However, the role of p53 is not explained, and the connection between this result and the main theme of the article is unclear. Further clarification is needed on why p53 was measured and how these results tie into the overall hypothesis.

4. Since both YY1 and TFAP2 are transcription factors, their overlapping binding motifs likely affect the expression of numerous genes. In Figure 3B, the statement that "these CBS potentially regulate cell junction assembly, cell-cell adhesion via plasma membrane adhesion molecules, and keratinocyte proliferation" seems speculative without detailed reasoning or evidence linking these pathways directly to the mechanisms of HPV-induced carcinogenesis. The experiment design of genome-wide screening of YY1 and TFAP2 competitive binding sites to find genes related to HPV-driven carcinogenesis seems inadequately substantiated.

5. Fig. 4A–B and Fig. S3 mention H3K27ac and H3K4Me3 marks in the promoter regions, but the manuscript briefly mentions these findings without explaining their significance or the impact they have on the study’s conclusions. More detail on these marks and their relevance to the research would be beneficial.

6. In Fig. 4G–H, the authors demonstrate that high levels of either PPP1R15B or LRRC37A correlate with a worse survival rate in cervical cancer patients. This experiment effectively supports the relevance of YY1 and TFAP2 competitive binding sites in cervical cancer and ties well into the manuscript’s central theme.

7. In the Discussion, the authors mention that "the residual TFAP2 binding does not repress transcription." This point is unclear and warrants further experimental verification or more in-depth discussion to clarify its implications.

Reviewer #3: The text of the manuscript needs editing for correct English language use throughout.

PLOS authors have the option to publish the peer review history of their article (what does this mean? ). If published, this will include your full peer review and any attached files.

**Do you want your identity to be public for this peer review?** For information about this choice, including consent withdrawal, please see our Privacy Policy .

Reviewer #1: No

Reviewer #2: No

Reviewer #3: No

**Figure resubmission:**

**Reproducibility:**



---

## [Decision Letter · Decision Letter 1]

24 Aug 2025

PPATHOGENS-D-25-00313R1

A genome-wide analysis of YY1 and TFAP2 competition on overlapping motifs reveals their roles in HPV-induced carcinogenesis

PLOS Pathogens

Dear Dr. Liu,

Thank you for submitting your manuscript to PLOS Pathogens. After careful consideration, we feel that it has merit but does not fully meet PLOS Pathogens's publication criteria as it currently stands. Therefore, we invite you to submit a revised version of the manuscript that addresses the points raised during the review process.

Please submit your revised manuscript within 30 days Oct 23 2025 11:59PM. If you will need more time than this to complete your revisions, please reply to this message or contact the journal office at plospathogens@plos.org. Please include the following items when submitting your revised manuscript:

We look forward to receiving your revised manuscript.

Kind regards,

Cary A. Moody

Academic Editor

PLOS Pathogens

Blossom Damania

Section Editor

PLOS Pathogens

Sumita Bhaduri-McIntosh

Editor-in-Chief

PLOS Pathogens

orcid.org/0000-0003-2946-9497

Michael Malim

Editor-in-Chief

PLOS Pathogens

orcid.org/0000-0002-7699-2064

**Journal Requirements:**

1) We do not publish any copyright or trademark symbols that usually accompany proprietary names, eg ©,  ®, or TM  (e.g. next to drug or reagent names). Therefore please remove all instances of trademark/copyright symbols throughout the text, including:

- TM on page: 16.

2) Please amend your detailed Financial Disclosure statement. This is published with the article. It must therefore be completed in full sentences and contain the exact wording you wish to be published.

2) State what role the funders took in the study. If the funders had no role in your study, please state: "The funders had no role in study design, data collection and analysis, decision to publish, or preparation of the manuscript.".

3) Please ensure that the figures are uploaded in a correct numerical order in the online submission form.

**Comments to the Authors:**

**Please note that one review is uploaded as an attachment.**

**Reviewers' Comments:**

Reviewer's Responses to Questions

**Part I - Summary**

Reviewer #1: The authors have correctly addressed all of my comments.

Reviewer #2: This report investigates the regulatory mechanism of the long non-coding RNA lnc-FANCI-2 in HPV-driven cervical carcinogenesis. While the manuscript provides novel insights into the antagonistic roles of YY1 and TFAP2 in regulating lnc-FANCI-2 and broader transcriptional networks, the manuscript has several shortcomings need to be addressed.

the writing lacks logical coherence, making it difficult for readers to follow the study’s flow. The rationale behind each experiment is not clearly articulated, nor is it easy to understand how specific conclusions are derived from the corresponding results. The manuscript would benefit greatly from improved organization and enhanced clarity to increase its readability.

Second, the logical framework of the manuscript is incomplete, particularly concerning the relationship between E7, YY1, and TFAP2. Although the authors express an intention to explore how E7 manipulates YY1 to drive HPV-induced transcriptional reprogramming, the manuscript lacks regulatory data directly linking E7 and YY1. This omission weakens the clarity and focus of the study's central objective.

Third, the biological relevance of the findings remains unsubstantiated. There are no experiments demonstrating that the proposed mechanism has a functional impact on HPV-induced tumorigenesis, which significantly diminishes the importance and credibility of the study.

Overall, this study provides compelling evidence of competitive binding between YY1 and TFAP2 at the lnc-FANCI-2 promoter through a series of well-executed experiments. With appropriate revisions to strengthen the logical and evidentiary framework of the manuscript, I would recommend this manuscript for acceptance.

Reviewer #3: This is a resubmitted manuscript. The authors have taken on board all of my suggestions from the first round of reviews and I think the manuscript is much improved.

**Part II – Major Issues: Key Experiments Required for Acceptance**

Reviewer #1: N/A

Reviewer #2: 1. The current study is limited to cell-based experiments. It is strongly recommended that the authors complement their findings with in vitro luciferase reporter assays to further validate the competitive binding of YY1 and TFAP2 to the lnc-FANCI-2 promoter.

2. In Figure 3E, the authors are encouraged to perform siYY1 in CaSki cells to determine whether TFAP2A binding increases upon YY1 knockdown and how this affects downstream lnc-FANCI-2 transcription. Conversely, YY1 overexpression in HFK cells should be conducted to assess whether TFAP2A binding to the lnc-FANCI-2 promoter is reduced and whether this impacts its transcription. These experiments would strengthen the mechanistic conclusions.

Reviewer #3: On reading the manuscript again, I now note that the ChIP experiments presented in Fig 3 are actually end point PCR and the data obtained from densitometry of PCR products separated by electrophoresis. I do not find the gel images anywhere in the manuscript. Perhaps these should be submitted as supplementary figures? Perhaps more worrying is that analysis of ChIP by end point PCR/densitometry is notoriously inaccurate. qPCR is a much better method for quantifying ChIP experiments and I suggest the authors consider reanalyzing their data in this way before the manuscript is accepted for publication.

**Part III – Minor Issues: Editorial and Data Presentation Modifications**

Reviewer #1: N/A

Reviewer #2: 1. The line and bar graphs throughout the manuscript do not include statistical significance analyses. Proper statistical validation is essential to support the conclusions drawn from these data.

Reviewer #3: (No Response)

PLOS authors have the option to publish the peer review history of their article (what does this mean? ). If published, this will include your full peer review and any attached files.

**Do you want your identity to be public for this peer review?** For information about this choice, including consent withdrawal, please see our Privacy Policy .

Reviewer #1: No

Reviewer #2: No

Reviewer #3: No

**Figure resubmission:**

**Reproducibility:**



---

## [Editor Report · Decision Letter 2]

8 Sep 2025

Dear Dr. Liu,

We are pleased to inform you that your manuscript 'A genome-wide analysis of YY1 and TFAP2 competition on overlapping motifs reveals their roles in HPV-induced carcinogenesis' has been provisionally accepted for publication in PLOS Pathogens.

Best regards,

Cary A. Moody

Academic Editor

PLOS Pathogens

Blossom Damania

Section Editor

PLOS Pathogens

Sumita Bhaduri-McIntosh

Editor-in-Chief

PLOS Pathogens

orcid.org/0000-0003-2946-9497

Michael Malim

Editor-in-Chief

PLOS Pathogens

orcid.org/0000-0002-7699-2064
---

## [Editor Report · Acceptance letter]

Dear Dr. Liu,

We are delighted to inform you that your manuscript, "A genome-wide analysis of YY1 and TFAP2 competition on overlapping motifs reveals their roles in HPV-induced carcinogenesis," has been formally accepted for publication in PLOS Pathogens.

Best regards,

Sumita Bhaduri-McIntosh

Editor-in-Chief

PLOS Pathogens

orcid.org/0000-0003-2946-9497

Michael Malim

Editor-in-Chief

PLOS Pathogens

orcid.org/0000-0002-7699-2064